# Potential Benefits of Daytime Naps on Consecutive Days for Motor Adaptation Learning

**Yusuke Murata** [1,2], **Masaki Nishida** [1,2,*], **Atsushi Ichinose** [1,2], **Shutaro Suyama** [1,2], **Sumi Youn** [1,2] and **Kohei Shioda** [2,3]

1. Faculty of Sport Sciences, Waseda University, Saitama 359-1192, Japan
2. Sleep Research Institute, Waseda University, Tokyo 169-8050, Japan
3. Faculty of Human Sciences, Kanazawa Seiryo University, Kanazawa 920-8620, Japan
* Correspondence: nishida@waseda.jp; Tel.: +81-4-2947-6771

**Abstract:** Daytime napping offers benefits for motor memory learning and is used as a habitual countermeasure to improve daytime functioning. A single nap has been shown to ameliorate motor memory learning, although the effect of consecutive napping on motor memory consolidation remains unclear. This study aimed to explore the effect of daytime napping over multiple days on motor memory learning. Twenty university students were divided into a napping group and no-nap (awake) group. The napping group performed motor adaption tasks before and after napping for three consecutive days, whereas the no-nap group performed the task on a similar time schedule as the napping group. A subsequent retest was conducted one week after the end of the intervention. Significant differences were observed only for speed at 30 degrees to complete the retention task, which was significantly faster in the napping group than in the awake group. No significant consolidation effects over the three consecutive nap intervention periods were confirmed. Due to the limitations of the different experimental environments of the napping and the control group, the current results warrant further investigation to assess whether consecutive napping may benefit motor memory learning, which is specific to speed.

**Keywords:** nap; sleep; motor adaptation; learning; consolidation

## 1. Introduction

Sleep has been considered a crucial post-practice activity for the evolution of memory. Numerous studies have demonstrated the beneficial role of sleep in memory consolidation, under a process in which the newly learned information may be organized and reactivated offline, such that initially labile memory traces become more robust and fixed [1,2]. Among different arrays of memory systems, declarative memory was positively associated with sleep-dependent consolidation [3]. Subsequently, this concept has extended to other forms of memory, such as the procedural memory domain. In early studies, sleep-dependent memory processing was observed in the domains of declarative and procedural memory [4,5]. Accumulating evidence has indicated that motor memory consolidation refers to the "off-line" state in sleep, wherein memory traces are presumably consolidated in a different manner than on-line [6].

Procedural memory is defined as a collection of abilities to acquire various skills that do not involve the direct recall of previous episodes [7]. Motor learning includes both simple and complex motor skill behaviors from daily activities, such as household chores, to techniques involving sports and music. Nevertheless, motor memory processing may depend on the nature of the motor learning demands. Adaptation to visuomotor rotations is one of the most widely studied paradigms of motor learning skills, in which individuals have to adapt to sensorimotor perturbations (motor adaptation, MA) [8]. MA represents the capacity to modify motor behavior in response to changes in the environment [9]. Plihal and

Born have initially demonstrated that sleep following an initial session of MA increased performance in a subsequent retest by employing a mirror tracing task [4]. Hereafter, more studies have replicated the beneficial effect of sleep on MA with different variations of tasks and situations [10–15]. In the recently employed gross MA task, it has been reported that sleep has a facilitative effect on skill acquisition and consolidation [12,16]. However, a recent meta-analysis questioned the previous findings that sleep facilitates MA learning [17]. Debas et al. reported that sleep had no effect on MA performance, although sleep significantly improved motor sequence learning (MSL) performance [18]. A greater effect on MA consolidation is induced by practice and not by sleep [19]. However, whether MA consolidation are mediated by sleep still remains controversial.

Daytime napping is regarded as a healthy habit by reducing sleepiness resulting in refreshment in the afternoon, as well as an effective countermeasure to sleep deprivation [20]. While napping has a prominent benefit on motor learning tasks, the beneficial effect was demonstrated predominantly in an MSL task [21,22]. In contrast to an MSL task, a few studies have explored the association between MA consolidation and napping. Backhaus et al. have found no significant differences in the changes of MA performance between the three groups (awake, short-nap, and long-nap groups), in retesting the following day [11]. In addition, Hoedlmoser et al. have demonstrated the inverse effect of napping on MA consolidation, provided the decreased performance after taking a nap [22]. However, these studies investigating the association between motor leaning and napping examined the effect of napping on a single occasion. Napping not only helped reduce daytime sleepiness but also played an essential role in recovering from a chronic loss of sleep, resulting in an extension of overall sleep time [1]. Sleep extension has been shown to improve visuospatial processing [21,23] and sports performance requiring motor procedural learning [24–26]. The improvement in performance due to extended sleep may be the result of eliminating "sleep debt," which is considered chronic sleep loss [27]. Although consecutive napping may be useful for reducing cognitive deficit due to sleep deprivation, no studies have examined motor memory consolidation through consecutive napping.

Therefore, this study aimed to explore motor learning employed when performing MA tasks by analyzing three-day consecutive daytime napping. To examine the effect of consecutive napping opportunity on MA consolidation, the retest was performed one week after the end of the three-day napping intervention. Polysomnography was recorded to score the sleep stages and examine the association between the magnitude of consolidation and sleep architecture. We hypothesized that there would be a significant consolidation effect in consecutive napping opportunity; this would confirm that the habit of napping is practical for overcoming sleep deprivation. Our findings provide experimental evidence for the effects of consecutive napping habits on motor memory function.

## 2. Results

### 2.1. Sleep Variables at Night during the Experiment

Nighttime sleep variables measured using the waist actigraphy during the RAT experiment period (DAY 1 to DAY 3) were calculated for the napping group. The sleep duration was 371.0 min (SD 83.0 min), and the sleep efficiency was 66.0% (SD 16.7%). Regarding nighttime sleep measurements during the entire experimental period from DAY 1 to DAY 11, the data of four participants were excluded due to poor recording. The mean sleep duration was 365.0 min (SD 95.0 min), and the sleep efficiency was 67.7% (SD 11.7%). These data indicate that there was no significant sleep deprivation over the duration of the experiment.

### 2.2. Nap Propensities

Table 1 summarizes the sleep variables during napping on each experimental day. No significant differences were obtained in each variable among days. The correlation analysis between sleep variables and MA indices showed no significant correlation, by using FDR methods to verify multiple correction.

**Table 1.** Sleep parameters during napping on each experimental day.

| | TIB (min) | TST (min) | SE (%) | SL (min) | Wake (min) | N1 (min) | N2 (min) | N3 (min) |
|---|---|---|---|---|---|---|---|---|
| DAY 1 | 50.7 (4.3) | 39.8 (5.0) | 78.4 (1.6) | 4.9 (1.7) | 10.9 (3.6) | 14.0 (7.1) | 18.2 (9.1) | 7.6(9.2) |
| DAY 2 | 49.7 (2.8) | 39.4 (7.6) | 79.1 (1.3) | 4.1 (1.5) | 10.2 (6.3) | 11.6 (3.7) | 18.5 (10.6) | 9.4 (9.4) |
| DAY 3 | 52.7 (3.0) | 37.5 (12.1) | 70.9 (1.3) | 4.0 (1.6) | 15.3 (11.3) | 12.4 (8.6) | 15.5 (9.1) | 9.5 (8.7) |
| Average | 51.0 (2.2) | 38.9 (5.0) | 76.2 (0.8) | 4.3 (1.1) | 12.7 (3.6) | 12.7(3.6) | 17.4 (5.8) | 8.8 (7.6) |

Abbreviations: TIB, total time in bed; TST, total time of sleep; SE, sleep efficiency; SL, latency to sleep onset; N1, NREM sleep stage 1; N2, NREM sleep stage 2; N3, NREM sleep stage 3. Sleepiness scale values are shown as means (standard deviations).

No statistical differences were obtained through the three-day experiment.

*2.3. Subjective Sleepiness*

The changes in subjective sleepiness in the napping group are shown in Table 2. Sleepiness after napping was significantly reduced compared with sleepiness before napping on each experimental day. The mean value of sleep quality evaluated using the visual analog scale during the nap was better than that during the median point, indicating that the subjective napping was comfortable.

**Table 2.** Changes in subjective sleepiness before and after napping.

| | KSS (pre Nap) | KSS (Post Nap) | Degree of Improvement in KSS | VAS |
|---|---|---|---|---|
| DAY 1 | 5.3 (1.1) | 3.7 (1.4) | 1.7 (1.6) | 4.9 (1.7) |
| DAY 2 | 5.3 (1.3) | 3.7 (1.5) | 1.7 (1.3) | 4.1 (1.5) |
| DAY 3 | 4.7 (1.5) | 3.4 (2.0) | 1.2 (1.3) | 4.0 (1.6) |
| Average | 5.1 (0.9) | 3.6 (1.2) | 1.5 (0.8) | 4.3 (1.1) |

Abbreviations: KSS, Karolinska sleepiness scale; VAS, visual analog scale (for self-rated sleep quality, 0—poor, 10—good). Values are shown as means (standard deviations).

*2.4. Required Time*

The results of total angle time required at each measurement point for the napping and awake groups are shown in Figure 1.

For overall angles combined, the main effect of the group was not significant ($F_{(1, 16)} = 0.120$, $p = 0.734$, $\eta_p^2 = 0.001$). However, a significant main effect by measurement point was obtained ($F_{(2.46, 39.40)} = 50.62$, $p < 0.001$, $\eta_p^2 = 0.766$). Divided by each angle, the main effect by measurement point was significant for each angle (30° $F_{(2.28, 36.55)} = 19.68$, $p < 0.001$, $\eta_p^2 = 0.557$; 60° $F_{(2.34, 37.51)} = 36.41$, $p < 0.001$, $\eta_p^2 = 0.701$; 90° $F_{(3.40, 54.43)} = 50.36$, $p < 0.001$, $\eta_p^2 = 0.756$). No significant interaction between the groups and the measurement point ($F_{(2.46, 39.40)} = 1.130$, $p = 0.342$, $\eta_p^2 = 0.068$) was observed.

Although post-hoc analysis revealed no significant differences among measurement points, the speed of the final RT for 30° was significantly faster in the napping group compared to the awake group ($p = 0.034$, Cohen's d = 0.853).

Similarly, a percentage increase of speed from D4 to the final RT was significantly greater in the awake group compared to the napping group ($p = 0.021$, Cohen's d = 0.861). No significant percentage changes were observed in other angles.

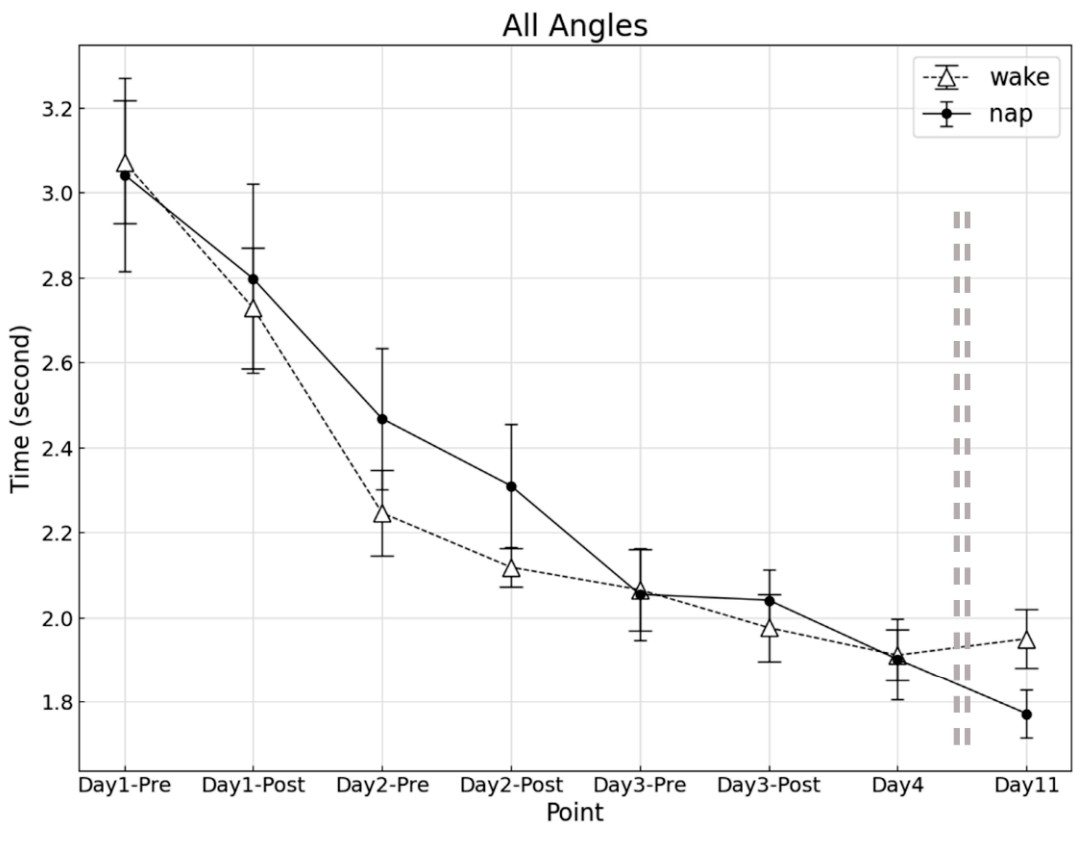

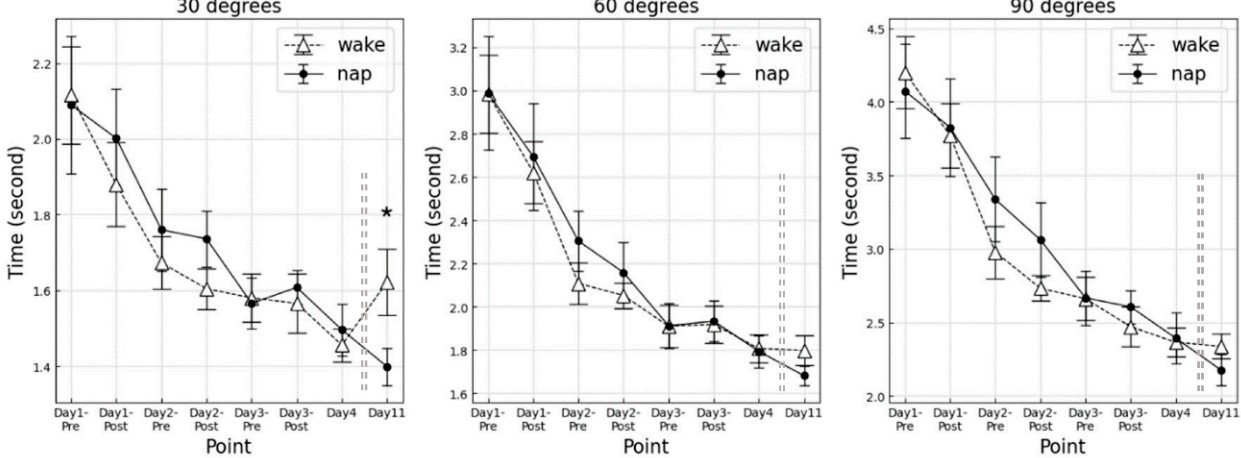

**Figure 1.** Time required at each measurement point. Upper, averaged angle; Lower left, 30°; middle, 60°; right, 90°. Error bars mean standard error. * $p < 0.05$.

### 2.5. Trajectory Length

The results of the trajectory length at each measurement point for the napping and awake groups are shown in Figure 2.

For overall angles combined, neither the main effect of group (F (1, 16) = 2.650, $p = 0.123$, $\eta_p^2 = 0.066$), nor the main effect by measurement point (F (2.51, 40.21) = 2.570, $p = 0.077$, $\eta_p^2 = 0.137$) was found to be significant. No significant interaction was observed between group and measurement point (F (2.51, 40.21) = 0.44, $p = 0.689$, $\eta_p^2 = 0.024$).

When divided by angles, the main effect by measurement point was significant for each angle (30° F (3.12, 49.91) = 6.46, $p = 0.001$, $\eta_p^2 = 0.282$; 60° F (3.02, 48.38) = 4.11, $p = 0.011$, $\eta_p^2 = 0.210$; 90° F (7112) = 2.27, $p = 0.034$, $\eta_p^2 = 0.124$). Post-hoc analysis showed no significant differences among measurement points. The final RT was not significantly

different between the two groups for each angle. Comparison of a percentage change of trajectory length from D4 to the final RT showed no significant differences between the groups for each angle.

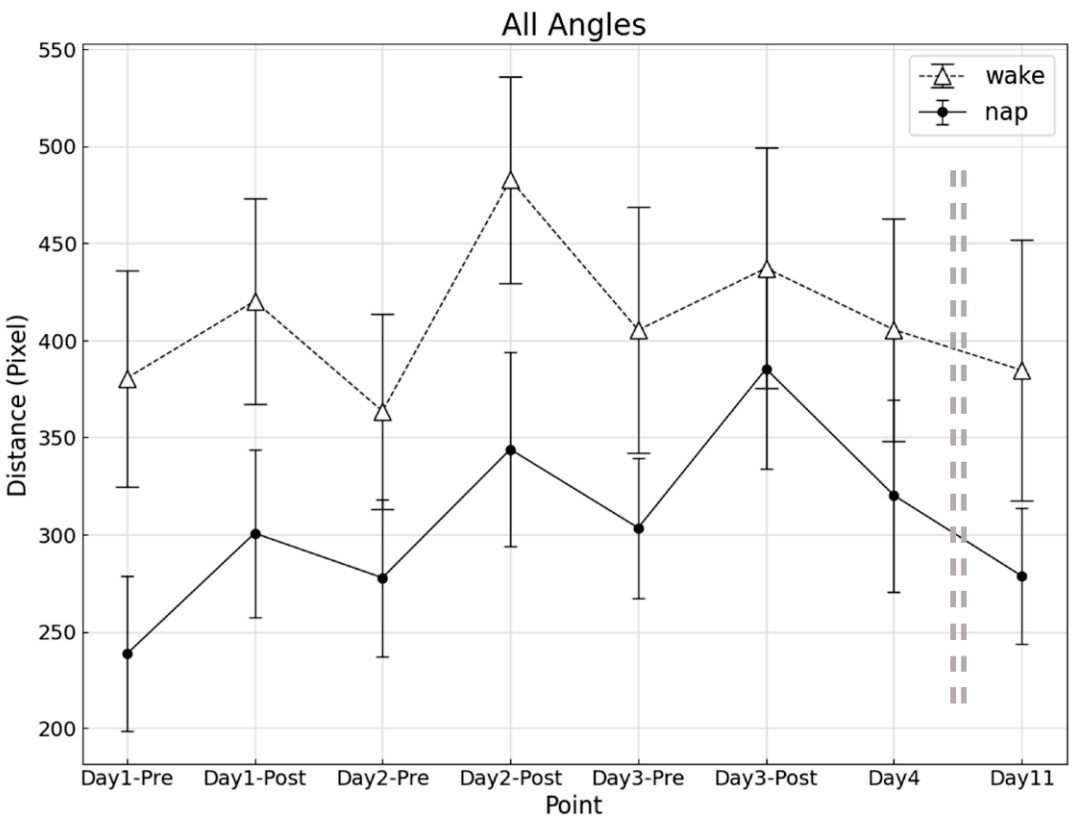

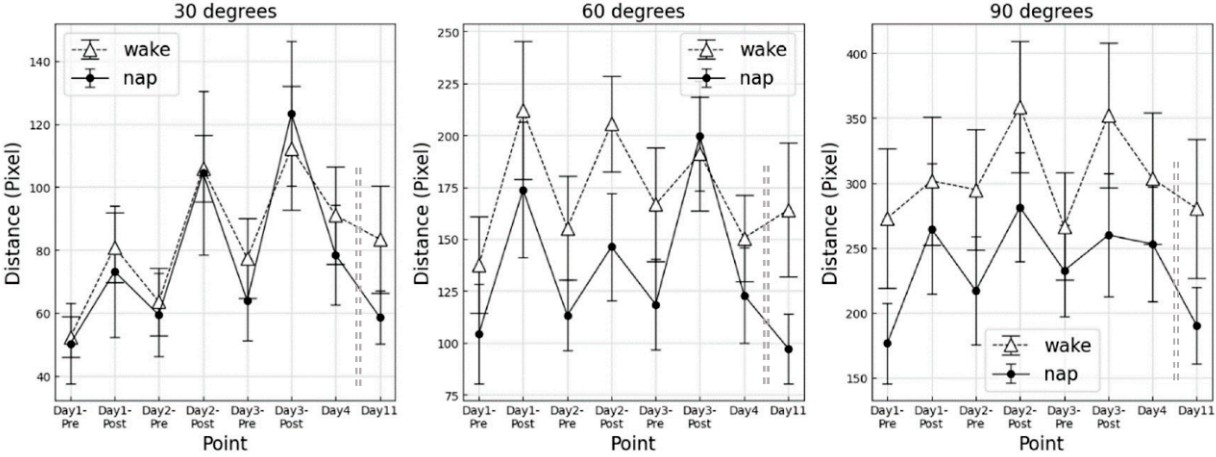

**Figure 2.** Trajectory length at each measurement point. Upper, averaged angle; Lower left, 30°; middle, 60°; right, 90°. Error bars mean standard error.

### 2.6. Synthetic Index (Required Time × Trajectory Length)

The results of the synthetic index at each measurement point for the napping and awake groups are shown in Figure 3.

For overall angles combined, a significant main effect of group was obtained (F (1, 16) = 4.993, $p$ = 0.040, $\eta_p^2$ = 0.098), as well as a significant main effect by measurement point (F (2.89, 46.20) = 6.01, $p$ = 0.002, $\eta_p^2$ = 0.264). No significant interaction was observed between group and measurement point (F (2.89, 46.20) = 1.75, $p$ = 0.172, $\eta_p^2$ = 0.100). In the

RT trial, the napping group had a shorter required time than the awake group ($p = 0.241$, $d = 0.130$), although no significant difference was observed.

Divided by each angle, a significant main effect of group was obtained in the 60° angle trial (30° F (1, 16) = 3.196, $p = 0.090$, $\eta_p^2 = 0.066$; 60° F (1, 16) = 7.471, $p = 0.015$, $\eta_p^2 = 0.097$; 90° F (1, 16) = 3.700, $p = 0.072$, $\eta_p^2 = 0.104$). The significant main effects of group by measurement were acquired in all angle deviations (30° F (3.28, 52.54) = 3.71, $p = 0.014$, $\eta_p^2 = 0.042$; 60° F (3.10, 49.63) = 7.09, $p < 0.001$, $\eta_p^2 = 0.072$; 90° F (3.27, 52.26) = 5.46, $p = 0.002$, $\eta_p^2 = 0.091$). No significant interactions were observed between group and measurement point for each angle (30° F (3.28, 52.54) = 0.75, $p = 0.538$, $\eta_p^2 = 0.042$; 60° F (3.10, 49.63) = 1.60, $p = 0.201$, $\eta_p^2 = 0.072$; 90° F (3.27, 52.26) = 1.41, $p = 0.247$, $\eta2 = 0.091$).

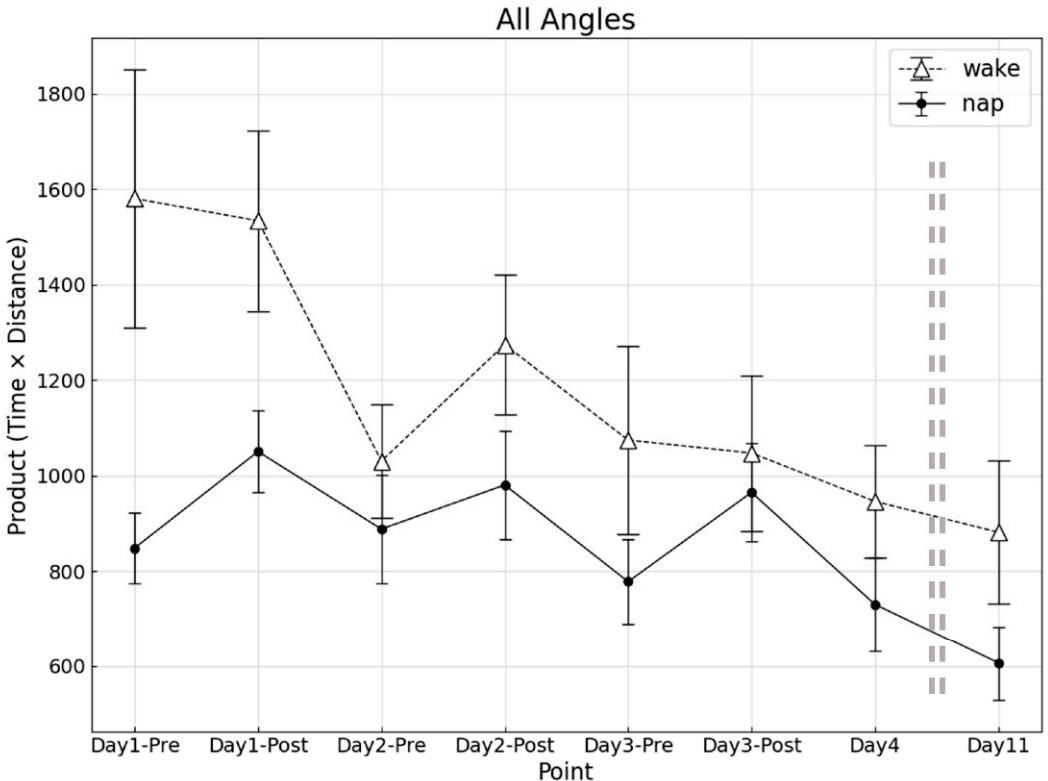

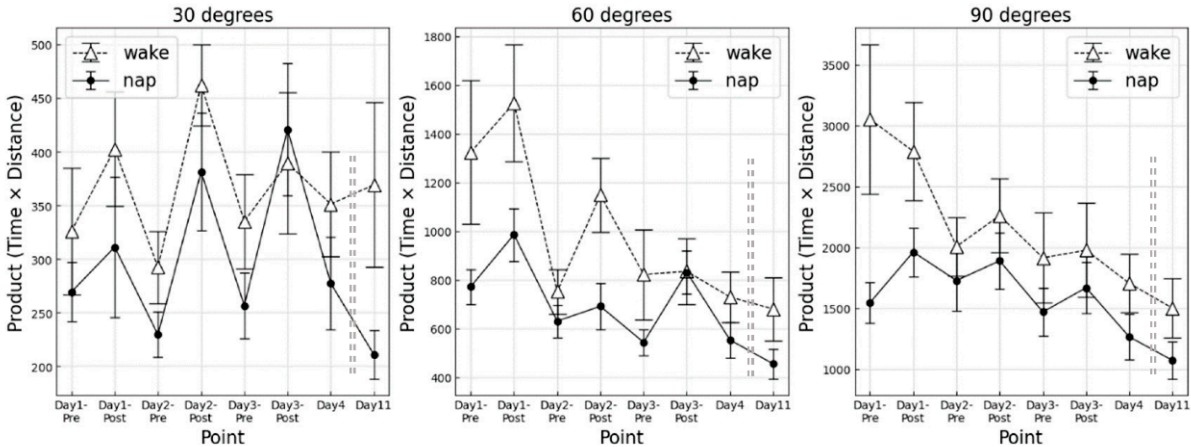

**Figure 3.** Trajectory length at each measurement point. Upper, averaged angle; Lower left, 30°; middle, 60°; right, 90°. Error bars mean standard error.

Post-hoc analysis showed no significant differences among measurement points. The final RT also revealed no significant difference between the two groups for each angle.

Additionally, a comparison of a percentage change of synthetic index from D4 to the final RT revealed no significant differences between the groups for each angle.

## 3. Discussion

In the current study, we analyzed the effects of three-day consecutive daytime napping on MA learning and the subsequent consolidation of motor memory by combining behavioral data with contemporaneous EEG recordings. Variables, including speed, accuracy, and a combined speed–accuracy index, showed significant main effects in the napping group compared with the awake group. However, the significant consolidation effect compared to the retest session was found only in the speed of the 30° angle in the napping group compared with the awake group. While the current results are unlikely to demonstrate the effectiveness of consecutive napping on motor adaptation learning, they also suggest a possibility for speeds under less complex conditions.

Although motor learning tasks have a distinct nap benefit [28–31], evidence that napping has an incremental effect on MA consolidation has been limited. The first formative study by Plihal and Born revealed that the MA facilitation after sleep was specific to the latter part of night sleep, consisting of a higher proportion of REM sleep [4]. A facilitative effect of nocturnal sleep that includes sufficient REM sleep on subsequent MA consolidation has been consistently reproduced [10,13,14,32,33]. Considering that napping contains less REM sleep than nocturnal sleep, daytime napping is unlikely to enhance MA learning. Meanwhile, several studies have reported that even whole night sleep provided no overnight improvement of MA compared with the wake control group [11,34–36]. Whether sleep facilitates MA consolidation remains under debate; however, consecutive napping may have possibilities for the consolidation of the MA learning process in specific conditions because of a lower capacity to maintain the memory trace for MA during wakefulness, requiring a subsequent time window for the stabilization of the acquired MA skill [10,13,14,32,33]. It should be noted that the effects of consecutive naps on memory consolidation may not be obvious during the intervention period.

While the effect of napping on MA consolidation appears inconsistent, consecutive daily napping is considered a practical strategy to increase sleep duration, which may be equivalent to extended sleep [24]. Although the participants of both the napping and awake groups reported that they kept a regular sufficient sleep duration, a past study demonstrated that even habitual 6 h sleep, which is considered a normal sleep duration, produced cognitive performance deficits equivalent to up to two nights of total sleep deprivation [37]. Extended or additional sleep has the effect of reducing subsequent sleepiness, as well as the addition of active off-line memory consolidation. Subsequent sleepiness after napping was reduced in this experiment, suggesting the possibility of improved performance due to the increased vigilance involved in reduced sleepiness. However, the previous study postulated the potential lack of association between subjective measures of sleepiness and actual performance, representing the less pivotal of these scales for sleep-dependent memory consolidation [37]. Apart from the sleepiness, additional consolidation process also occurred, presumably during slow-wave sleep. Poor quantity and quality of sleep induced by slow-wave sleep deprivation have been shown to impair the sleep-related consolidation of a visuomotor adaptation task [38]. Regarding compensation for sleep loss, a previous study has demonstrated that sleep duration was considerably unaffected by whether the sleep was situated nocturnally or split between nocturnal anchor sleep periods and daytime naps [39,40]. According to the notion that daytime napping has the potential to work as a supplement against unperceived sleep loss, consecutive napping may presumably provide a similar effect to that of sleep extension on procedural motor learning.

In this experiment setting, speed was facilitated more from napping opportunity than trajectory length, resulting in a beneficial effect on RT in the relatively less complex condition of 30°. The present results are unlikely to be consistent with previous research that differentiated the effects of training in speed and the accuracy components of motor

tasks, with speed benefiting most from training [41]. One possibility as to why speed was significantly enhanced is that the participants may have prioritized only speed for the cursor movement, making trajectory length a secondary consideration. The participants were requested to "move the cursor to the target as fast and as straight as possible," implying that the participants prioritized speed since it was written at the beginning of the warning. In addition, however, the facilitating effect on the retest was observed in the relatively easy 30° rather than the more difficult 60° and 90° conditions, suggesting that task difficulties in MA learning are related to gaze control. A past study demonstrated that gaze control during a visuo-adaptation task was modified more in the 30° condition compared to the 75° condition [42], implying that the sleep-dependent learning of gaze control would be exhibited in an easier task, such as the 30° condition.

In terms of trajectory length, the awake group required an overall longer trajectory length compared with the napping group. The trajectory length of the awake group was generally higher than that of the napping group, which may be partly due to the experimental environment. The experiment in the awake group was conducted at home, possibly resulting in distraction from the experiment. Considering that the internet-based experiment has recently become a well validated alternative to traditional laboratory-based assessment, recent studies have indicated the inconsistency of negligible differences between the domestic and laboratory settings [43–45]. Herein, considering the similar tendency of the synthetic index, the awake group may have been less focused on the experiment than the napping group.

Neural networks have been implicated in the sleep-dependent learning of MA. In contrast to MSL tasks involving greater contributions from the cortico-striatal system, MA tasks primarily recruit a network of cortico-cerebellar structures instead [17]. A previous neuroimaging study revealed no significant difference in cerebral activities in the overnight intervention group compared with the wake control group [18]. Albouy et al. demonstrated that sleep deprivation after an initial adaptation session impaired learning, with increased activation in cerebello-cortical networks afterward [10]. This result suggests that the process of MA consolidation predominantly occurs in earlier learning stages within the brain system related to motor coordination. Thus, daily napping after MA learning possibly helps memory consolidation further after a certain amount of time scale involving the neural networks of the cortices and cerebellum.

Converging evidence has demonstrated that MSL consolidation was robustly associated with sleep spindle, regarded as a hallmark of NREM sleep [30,46]. The sleep-dependent memory consolidation of MA was shown to be related to increased regional slow-wave activity during NREM sleep in the task-relevant region [13]. Aside from slow-wave sleep, fast spindle activity was associated with MA in a mirror-tracing task [47]. This inconsistency is in line with our findings that revealed no association between sleep architecture and the extent of MA learning. While REM sleep density may be involved in MA learning [48], the effect of sleep propensity on memory consolidation remains unclear, and further studies are needed to explore the role of subsequent sleep in MA learning, including temporal protection against relevance.

Nevertheless, the present study has several limitations. First, the sample size was insufficient to demonstrate the incremental effect of napping, not providing support for a benefit of repeated napping after the nap intervention. Benefits should be demonstrated with larger samples and a within-subject design with more controlled protocols. Second, the duration of the napping intervention was limited to three days, which is relatively inadequate to confirm the effect of napping on motor learning. By extending the intervention period, the effect on motor learning could be verified. Third, the environment of the experiment was different in different groups. Since the experiments in the napping group were conducted in the laboratory, while those of the awake group were conducted in a place other than the laboratory, the degree of concentration during the experiments differed, which may have affected the results of the trajectory length. The adequacy of the results of the motor adaptation learning should be tested in a similar experimental environment.

Lastly, only the required time was displayed on the screen, resulting in attenuated results of the trajectory length.

## 4. Materials and Methods

### 4.1. Participants

Participants were recruited through posters and web-based advertisements in the university. Participants were screened by the Pittsburgh Sleep Quality Index (PSQI) to assess overall sleep quality [49] and by a morningness–eveningness questionnaire (MEQ) to evaluate circadian inclination [50]. Inclusion criteria: 1. participants' ages ranged from 18 to 30 years old; 2. subjective scale evaluation included the following: PSQI < 10 points, MEQ scores were >30 points <70 points; 3. participants have never been diagnosed with drug or alcohol abuse, neurological, psychiatric, or sleep disorders; 4. Non-habitual nappers who take regular napping opportunity. Altogether, 20 male university students were assigned to either the napping group (*n* = 10, mean age: 23.1 ± 1.3 years) or awake group (*n* = 10, mean age: 22.2 ± 3.5 years). The participants were requested to maintain a regular sleep schedule 1 week prior to the experiment. The participants were also required to abstain from caffeine and alcohol throughout the study duration and to refrain from non-experimentally measured naps, confirmed by a post-experiment questionnaire. The examiner directly confirmed all conditions above by careful interview.

The sample size was calculated using G * Power 3.1.9.6 with an effect size determined by $\eta_p^2 = 0.06$, and the values for $\alpha$ and power were set at 0.05 and 0.80 for MA performance, respectively [51]. The required sample size was eight for each group to evaluate MA performance, plus two participants were required (i.e., 20 participants in total).

Based on the PSQI, the participants in both groups were observed to have proper sleeping habits (4.7 ± 1.9). The score MEQ (46.3 ± 7.2) indicated that the participants comprised the intermediate type, suggesting fewer deviations in the chronotype. The Academic Research Ethical Review Committee of Waseda University approved all activities (IRB #2019-193), and all participants provided informed consent. The study was conducted in accordance with the 1975 Declaration of Helsinki, as revised in 2013.

### 4.2. Experimental Designs: Napping and Awake Groups

The experimental procedure is illustrated in Figure 4. The experiment comprised a five-day schedule: Day 1 (D1), Day 2 (D2), Day 3 (D3), Day 4 (D4), and Day 11 as a retention test (RT). Both groups performed the first task at 13:00 (pre-nap), and the second task at 15:00 p.m. (post-nap), with a 2 h interval. The nap group performed the experimental task in the laboratory, whereas the no-nap group performed the task while remaining awake at a similar time in their home environment. Because the home environment was used, the no-nap group was instructed by the experimenter (YM) through an online videoconference system via smartphone.

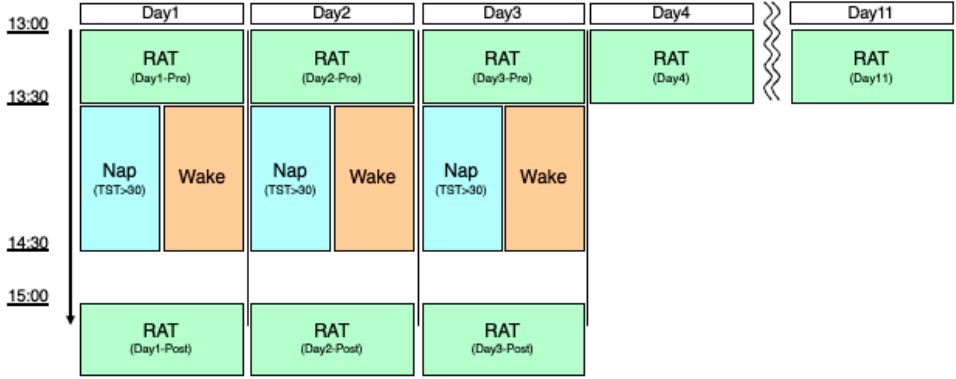

**Figure 4.** Schematic diagram of the experimental protocol. Abbreviations: RAT, rotation adaptation task; TST, total sleep time.

For the napping group, the experiment on D1, D2, and D3 started at 13:00 in the laboratory. The participants performed the pre-nap experimental tasks. After the completion of the pre-nap task, they answered the Karolinska sleepiness scale to assess subjective sleepiness. Subsequently, electrodes were attached to the scalp of the participants in the napping group to record sleep EEG during the nap. For the awake group, the experiment was conducted in the home environment. Similarly, beyond the experimentally recorded midday nap, those in the napping group were also instructed not to nap before or after the noon sleep session. The D4 and RT (D11) sessions were conducted once at 13:00.

After completing the task, the nap group entered a quiet, air-conditioned, dark room in the laboratory after being attached to the EEG for sleep measurement. Participants in the nap group sat deeply in a soft beanbag chair (Yogibo MAX, Webshark Inc., Osaka, Japan) and were requested to take a nap after the lights were turned off at 13:30. Then, they were woken up by an examiner (YM) at 14:30. Considering the decreased vigilance after wakeup (conceptualized as 'sleep inertia'), post nap tasks were conducted at 15:00, approximately 30 min after awakening. On the other hand, participants were instructed not to nap during the equivalent period (13:00–15:30); instead they were allowed to do passive activities (e.g., use computers or smartphones, watch television, play videogames, read, listen to music). The activities of participants during this period were monitored and confirmed online by the experimenter.

Both groups performed the experimental task at the following eight points: D1-pre, D1-post, D2-pre, D2-post, D3-pre, D3-post, D4, and RT (D11), with "pre" indicating the task before the equivalent timing of napping and "post" indicating the time after napping.

### 4.3. Sleep Measurements at Night

In the napping group, the participants were requested to an MTN-220 (Acos Co., Ltd., Nagano, Japan) on the front side of the trunk by clipping it to their waist belt or the edge of their trousers/pants. This actigraph is a small and light (9 g) coin-shaped device (external dimensions of 27 mm in diameter and 9.8 mm in depth, including the clip) that records the amount of physical activity by employing an internal three-axis accelerometer, enabling sleep evaluation at home prior to the experiment. The sensitivity and specificity of MTN-220 were equivalent to those determined for conventional actigraphy [52] and PSG [53]. MTN-220 devices generated sleep variables, including total sleep period time and sleep efficiency (i.e., the ratio of total sleep time over time in bed).

### 4.4. Rotation Adaptation Task

To investigate motor memory consolidation associated with consecutive napping, a rotation adaptation task (RAT) was adopted in the experiment task (Figure 5). The task consisted of eight white, circular target objects and one green, circular cursor object, placed on the circumference at equal angles (45° intervals). The cursor was placed at the center of the circle and could be moved. When the task started, one of the targets turned red, and the participant was asked to move the cursor to reach the red target (one trial). The cursor moved in the direction of a certain additional angle from the direction the participant moved it, with six additional angles: ±30°, ±60°, and ±90°. The additional angle was randomly determined for each trial and displayed on the screen during each trial. The participants were asked to instantly recognize the additional angle displayed on the screen for each trial and adapt the rotational effect of the additional angle to precisely bring the cursor to the target. After the cursor reached the target, the next trial started after a 1 s interval.

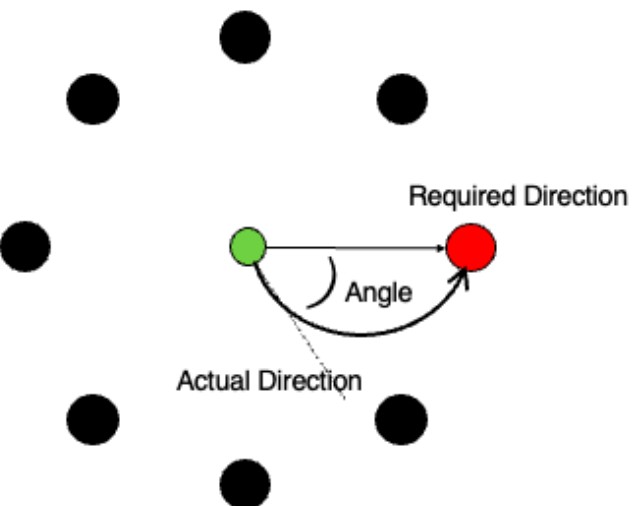

**Figure 5.** Schematic diagram of the rotation adaptation task (RAT).

The experimental tasks were performed on the screen of a laptop computer with a 15.6-inch (192 × 1080) display. A trackball mouse (KT-2337/72337 JP, Kensington, San Mateo, CA, USA) was used as the input interface to control the cursor. The participants were instructed to sit in a comfortable chair with the laptop PC placed on the desk in front of them. With the orientation of the trackball mouse and forearm aligned to face the front, the participants operated the trackball using the index and middle fingers. The distance between the center of the circle, which corresponds to the initial position and the center of each target, was 400 px. The size of the target itself was 40 px in diameter, and the cursor itself was 20 px in diameter.

### 4.5. PSG Recording and Sleep Stage Scoring

Polysomnography recording was performed during napping in accordance with standardized techniques [51], using digital EEG, electromyography (EMG), and electrooculography (EOG) signals acquired with a Polymate mini AP108 (Miyuki Giken Ltd., Tokyo, Japan). The sampling rate was at 256 Hz, and the high- and low-pass filters were 0.3 Hz and 35 Hz, respectively. A referenced PSG electrode montage was utilized, including EEG sites C4 and O2, EOG referenced to A1 and A2, and left and right outer canthi. Each sleep epoch of the PSG record was scored every 20 s by a clinical professional technologist blinded to the interventions according to the American Academy of Sleep Medicine manual scoring rules [52]. Although the 20 s epoch was shorter than that officially defined in the AASM scoring rule, applying a shorter epoch provides more accurate validity in determining sleep stages [54]. The signals were displayed on a computer monitor and rated visually, epoch by epoch, as either rapid eye movement (REM) sleep, non-REM (NREM) stages 1–3, awake, or movement time. Slow-wave sleep consisted of stage 3 NREM sleep.

### 4.6. RAT Data Analyses

The RAT analysis results consisted of three items: the time required for the cursor to reach the target (required time), the length of the trajectory required for the cursor to reach the target (trajectory length), and the value obtained by multiplying the above two items (synthetic index: required time × trajectory length). The required time indicated the smoothness of the cursor movement, and the trajectory length showed the distance loss when moving the cursor, implying the accuracy of the task. The synthetic indicator

indicated the combined results of the time required and the trajectory length. The analysis trajectory length *L* was calculated using the following formula:

$$L = \sum_{k=1}^{n} \sqrt{(x_k - x_{k-1})^2 + (y_k - y_{k-1})^2} - 400$$

where "*n*" indicated the number of frames from the start of the cursor movement until it reached the target, "*x*" the x-coordinate of the cursor, and "*y*" the y-coordinate of the cursor. To evaluate the loss from the shortest distance, the actual trajectory length minus 400 (px) was used as the trajectory length for analysis. Additionally, the trajectory length was adopted for analysis because it reflected the angle error more properly. The exclusion criteria were as follows: more than +2 standard deviations [SD] $\pm 90°$ in the D1-pre and D1-post for the time required and more than 2512 pixels for the trajectory length of one circumference of the target.

*4.7. Statistical Analysis*

The Shapiro–Wilk test showed that all data were normally distributed; thus, no transformation was required. For each of the 12 items, a two-way repeated measures analysis of variance (ANOVA) was conducted by group (napping and awake groups) × measurement point (D1-pre, D1-post, D2-pre, D2-post, D3-pre, D3-post, and D4). Greenhouse–Geisser correction was conducted to evaluate degrees of freedom as necessary. When the Mauchly test was significant, we adapted values with the Greenhouse–Geisser correction. To account for multiple comparisons, the ANOVA *p*-values were controlled using the false discovery rate (FDR) method [55]. When significant interaction effects were noted with the two-way ANOVA, analyses were broken down into paired *t*-tests as the post-hoc test. Statistical comparisons between the groups were independently performed using paired Student's *t*-tests. Effect sizes (Cohen's d) were calculated from *t*-statistics to reduce the probability of type II error [56]. A comparison of the percent change between D4 and D11 was performed to offset baseline differences caused by the experimental environment. Additionally, Pearson's correlation was used to examine the relationship between sleep components and MA learning indices. All analyses were performed using SPSS version 27 (IBM Corporation; Armonk, NY, USA).

## 5. Conclusions

Multi-day naps have limited possibilities to promote MA learning. While no significant interaction was observed between napping and speed, the napping group demonstrated a significantly faster speed than the awake group in the subsequent RT in less complex conditions. Neither a significant interaction nor main effects were confirmed for trajectory length or the combined index. Neither accuracy not a combined speed–accuracy index were significantly higher. These results suggest that consecutive napping possibly helps in MA consolidation representing the improved speed, which presumably stabilizes memory tracing after MA learning. Further research is warranted to duplicate the consolidation effect including the causal relationship between napping and MA consolidation. This will provide practical support in daily life activities, such as sports activities that require MA ability.

**Author Contributions:** Conceptualization, M.N., Y.M. and K.S.; methodology, M.N., Y.M. and K.S.; software, Y.M.; validation, M.N., Y.M. and K.S.; formal analysis, Y.M.; investigation, M.N. and Y.M.; resources, Y.M., A.I., S.S. and S.Y.; data curation, Y.M.; writing—original draft preparation, M.N. and Y.M.; writing—review and editing, M.N. and Y.M.; visualization, M.N. and Y.M.; supervision, M.N.; project administration, M.N.; funding acquisition, M.N. All authors have read and agreed to the published version of the manuscript.

**Funding:** This research was funded by JSPS KAKENHI [grant numbers JP18K10834, JP21K11458, and JP17H02139]. The funder had no role in the study design, data collection and analysis, decision to publish, or preparation of the manuscript.

**Institutional Review Board Statement:** The study was conducted in accordance with the Declaration of Helsinki of 1975, as revised in 2013, and approved by the Institutional Review Board of Waseda University on 20 August 2019 (IRB #2019-193).

**Informed Consent Statement:** Written informed consent was obtained from all participants involved in the study.

**Data Availability Statement:** The data that support the findings of this study are openly available at https://github.com/masakinishida/RotationAdaptation.git.

**Acknowledgments:** The authors would like to thank all participants in this study.

**Conflicts of Interest:** The authors declare no conflict of interest.

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
