# Peer review of "Potential Benefits of Daytime Naps on Consecutive Days for Motor Adaptation Learning"

_2624-5175, doi:10.3390/clockssleep4030033_

Round 1

Reviewer 1 Report (New Reviewer)

This paper explores the impact of daytime napping over three consecutive days on performance on a motor adaptation learning task in 20 healthy young male students (n=10 for the nap group and awake group, respectively). In my opinion, the study design presents a series of methodological flaws which make the results difficult to interpret in its current form.

As highlighted by the authors, acquisitions in the nap and awake groups were not performed in the same test environment (at home for the awake group – in the lab for the nap group).  Accordingly, it is mentioned for example in the discussion that the awake group required an overall longer trajectory length compared with the napping group, which may indicate that the groups were not equally focused on the task. Note that if this effect was statistically significant, it needs to be emphasized in the abstract and results sections.

Another concern is the absence of verified sleep-wake history during the days or week preceding the test sessions in both groups. Participants were advised to keep a regular sleep-wake schedule prior to the experiment, but if I understood correctly, adherence to pre-defined rest-activity cycles were not checked. It seems that it would have been technically feasible as Fitbits/Activity meters were used to assess night-time rest between Day1 and Day3 of the experiment (in the nap group only?). Also, while testing was performed at the same time of day for each participant, individual’s wake up time was not taken into account. Considering that the authors emphasize that “napping can be regarded as a healthy habit by reducing sleepiness resulting in refreshment in the afternoon, as well as an effective countermeasure to sleep deprivation”, it would have been important to gain information about sleep timing/duration/time since awake prior to the experimental session.

Earlier in the manuscript, it is mentioned that for nocturnal sleep during the experiment, the participants wore an activity meter (MTN-220, Acos. Ltd., Nagano, Japan) during nighttime sleep for the entire experimental period. It is not clear what was extracted from that device and how it complements the Fitbit assessment? Was MTN-220 measurement available for both groups?If so, it would be interesting to incorporate the results.

At first glance, it may appear odd that a sleep duration of 371.0 min (SD 83.0 min) and a sleep efficiency of 66.0% (SD 16.7%), as extracted from the FITBIT, indicates that there is no significant sleep deprivation over the duration of the experiment. Please underline this with appropriate references. 

Figures: what are the error bars reflecting?

Even though the context is different, no reference is done to studies revealing the impact of habitual daytime napping on night-time sleep and to the literature referring to napping as a health risk indicator - at least in the aged (see also recent publication by Leng et al. 2022). Accordingly, “daytime napping” can, but not necessarily has to be, regarded as a healthy habit. There is indeed a confusing use of the term “habitual napping” (where the volunteers habitual nappers? if so, why?) And its translation to a three-day nap intervention. In the same vain, it is not clear whether the authors believe that napping is beneficial for motor learning performance because of active consolidation processes taking place during nap sleep or because of napping reducing daytime sleepiness/as a countermeasure of sleep deprivation.

To my knowledge AASM sleep scoring rules suggest a 30-s window scoring and not a 20s as performed here. Also, I would recommend to report N1 and N2, separately.

The table indicates sleep efficiency per experimental day. The authors report that the duration and quality of napping on each experiment day are equivalent, which cannot be tested by inferential statistics (Day 1: 78.4%+-1.6;  Day 2 (79.1% +- 1.3); Day3: 70.9+- 1.3).

Even though interesting, the current results only partially allow to conclude that consecutive napping during MA learning enhances motor memory consolidation, since the study lacks a series of controls, particularly in the awake group (e.g. no sleepiness measure, no indication about prior sleep-wake history, no indication of sleep duration/timing during experimental days). Thus, are the observed effects reflecting differences in test environment, sleepiness, vigilance, sleep-wake history, enhanced consolidation processes or a combination of those?

Minor comments: In the discussion, it is mentioned that “the experiment in the awake group was not conducted at home”: I guess it is meant “was conducted at home”. p.2, line 54: learning instead of leaning.

Author Response

To Reviewer 1

This paper explores the impact of daytime napping over three consecutive days on performance on a motor adaptation learning task in 20 healthy young male students (n=10 for the nap group and awake group, respectively). In my opinion, the study design presents a series of methodological flaws which make the results difficult to interpret in its current form.

Response:

Thank you for your critical suggestion. According to your thoughtful comments, we revised the manuscript.

As highlighted by the authors, acquisitions in the nap and awake groups were not performed in the same test environment (at home for the awake group – in the lab for the nap group).  Accordingly, it is mentioned for example in the discussion that the awake group required an overall longer trajectory length compared with the napping group, which may indicate that the groups were not equally focused on the task. Note that if this effect was statistically significant, it needs to be emphasized in the abstract and results sections.

Response:

Your point is precisely the problem, and it was an inevitable limitation of the experimental environment due to the voluntary restraints imposed by COVID-19. 2-way AVOVA revealed a mix of conditions, some with significant main effects and some without statistical differences, including post-hoc. We have stated the description, but it is difficult to ascertain the causal relationship with experimental conditions. We have also added a note to the abstract and results to indicate that there are differences in experimental conditions that should be considered (Page 1, Line 24-25, Page7, Line 270-272).

Another concern is the absence of verified sleep-wake history during the days or week preceding the test sessions in both groups. Participants were advised to keep a regular sleep-wake schedule prior to the experiment, but if I understood correctly, adherence to pre-defined rest-activity cycles were not checked. It seems that it would have been technically feasible as Fitbits/Activity meters were used to assess night-time rest between Day1 and Day3 of the experiment (in the nap group only?). Also, while testing was performed at the same time of day for each participant, individual’s wake up time was not taken into account. Considering that the authors emphasize that “napping can be regarded as a healthy habit by reducing sleepiness resulting in refreshment in the afternoon, as well as an effective countermeasure to sleep deprivation”, it would have been important to gain information about sleep timing/duration/time since awake prior to the experimental session.

Response:

As your concern is right, the sleep-wake schedule prior to the start of the experimental intervention was just based on oral instructions, and was not actually measured by EEG or wearable devices. However, the experimenter confirmed the participants' compliance with the conditions prior to the experiment. Therefore, we assume that there were no major disruptions to the participants' sleep wake schedules at least on the day before the experiment. We added the description of the confirmation process in the main text (Page 2, Line 89- Page3, Line 99).

Additionally, not confirming the baseline of sleep-wake schedule prior to the experiment is indeed a deficiency. However, we did not enroll participants with obvious sleep-wake rhythm problems or those with considerably short or long sleep durations in the pre-experiment interviews.

Earlier in the manuscript, it is mentioned that for nocturnal sleep during the experiment, the participants wore an activity meter (MTN-220, Acos. Ltd., Nagano, Japan) during nighttime sleep for the entire experimental period. It is not clear what was extracted from that device and how it complements the Fitbit assessment? Was MTN-220 measurement available for both groups? If so, it would be interesting to incorporate the results.

Response:

Since we confirmed that nighttime sleep was not measured by MTN-220 but by Fitbit, we removed the relevant description.

At first glance, it may appear odd that a sleep duration of 371.0 min (SD 83.0 min) and a sleep efficiency of 66.0% (SD 16.7%), as extracted from the FITBIT, indicates that there is no significant sleep deprivation over the duration of the experiment. Please underline this with appropriate references. 

Response:

The data recorded by Fitbit showed that sleep duration is enough, but sleep efficiency looks lower than healthy population. Decreased quality of sleep may be attributed to daytime napping opportunity. We added the description to underpin the result (Page 6, Line 245-247).

Figures: what are the error bars reflecting?

Response:

Error bars means standard error. We defined it in the footnote of each Figure.

Even though the context is different, no reference is done to studies revealing the impact of habitual daytime napping on night-time sleep and to the literature referring to napping as a health risk indicator - at least in the aged (see also recent publication by Leng et al. 2022). Accordingly, “daytime napping” can, but not necessarily has to be, regarded as a healthy habit. There is indeed a confusing use of the term “habitual napping” (where the volunteers habitual nappers? if so, why?) And its translation to a three-day nap intervention. In the same vain, it is not clear whether the authors believe that napping is beneficial for motor learning performance because of active consolidation processes taking place during nap sleep or because of napping reducing daytime sleepiness/as a countermeasure of sleep deprivation.

To my knowledge AASM sleep scoring rules suggest a 30-s window scoring and not a 20s as performed here. Also, I would recommend to report N1 and N2, separately.

The table indicates sleep efficiency per experimental day. The authors report that the duration and quality of napping on each experiment day are equivalent, which cannot be tested by inferential statistics (Day 1: 78.4%+-1.6; Day 2 (79.1% +- 1.3); Day3: 70.9+- 1.3).

Response:

Although statistics revealed no significant differences among sleep propensities, addressing that the duration and quality of napping on each experiment day are equivalent may be an overstatement. The corresponding description has been deleted.

Even though interesting, the current results only partially allow to conclude that consecutive napping during MA learning enhances motor memory consolidation, since the study lacks a series of controls, particularly in the awake group (e.g. no sleepiness measure, no indication about prior sleep-wake history, no indication of sleep duration/timing during experimental days). Thus, are the observed effects reflecting differences in test environment, sleepiness, vigilance, sleep-wake history, enhanced consolidation processes or a combination of those?

Response:

Thank you for presenting the critical views. As you pointed out, the current results assumed partially that consecutive naps would be helpful for motor memory consolidation. My guess is that the observed effects are due to differences in the experimental environment, such as attention, concentration or alertness to surroundings. We addressed the caution in interpreting results in different experimental settings (Page 1, Line 24-25, Page7, Line 270-272).

Minor comments: In the discussion, it is mentioned that “the experiment in the awake group was not conducted at home”: I guess it is meant “was conducted at home”. p.2, line 54: learning instead of leaning.

Response:

Thank you for pointing out basic errors. We revised accordingly.

Reviewer 2 Report (New Reviewer)

  • How did the investigators select 20 male university students? Any selection bias?
  • How the investigators ascertain that those participants really took a nap?
  • The data of four participants were excluded due to poor recording. How did Fitbit activity provide poor recording?
  • What is the clinical implication?
  • What is the next step for future study?

Author Response

To Reviewer 2

How did the investigators select 20 male university students? Any selection bias?

Response:

Participants were initially recruited through posters and web-based advertisements in the university. We added the description to recruit the participants. We think that there is no arbitrary bias, although the bias of having recruited from our own university would be unavoidable.

How the investigators ascertain that those participants really took a nap?

Response:

Polysomnography (PSG) recording was performed during each napping opportunity, assuring that the participants was taking a nap. The description has been modified to show that the PSG was recorded during a nap.

The data of four participants were excluded due to poor recording. How did Fitbit activity provide poor recording?

Response:

Possibly the attachment to the wrist was loose and biological signal such as body movement was not detected properly, resulting in poor recording. This is an area for improvement in future experiments.

What is the clinical implication?

Exercise involving RAT, possibly in certain kind of sports, may have a facilitating effect on motor-skill improvement. The possibility of improving insufficient consolidation of learning due to sleep deprivation would be helpful for general population.

What is the next step for future study?

Number of athletes suffer from chronic sleep deprivation due to training, travel, and psychological stress. We would like to provide robust evidence that habitual napping can promote motor learning related to sports.

Round 2

Reviewer 1 Report (New Reviewer)

Globally, while a series of concerns were adressed in the response letter, other still remain open in my opinion. Rather minor changes have finally been performed in the manuscript. If I'm not mistaken, two of the concerns/comments were not addressed at all in the response to the reviewer document.

My concerns address the following answers:

Initial comment:

Considering that the authors emphasize that “napping can be regarded as a healthy habit by reducing sleepiness resulting in refreshment in the afternoon, as well as an effective countermeasure to sleep deprivation”, it would have been important to gain information about sleep timing/duration/time since awake prior to the experimental session.

Answer:

... the sleep-wake schedule prior to the start of the experimental intervention was just based on oral instructions, and was not actually measured by EEG or wearable devices. However, the experimenter confirmed the participants' compliance with the conditions prior to the experiment. Therefore, we assume that there were no major disruptions to the participants' sleep wake schedules at least on the day before the experiment. We added the description of the confirmation process in the main text (Page 2, Line 89- Page3, Line 99).

Additionally, not confirming the baseline of sleep-wake schedule prior to the experiment is indeed a deficiency. However, we did not enroll participants with obvious sleep-wake rhythm problems or those with considerably short or long sleep durations in the pre-experiment interviews.

Follow-up comment:

How exactly major disruption or its absence was assessed or quantified? Was there any information (even by questionnaire/interview) about prior sleep duration and timing or any other characteristic? In the same vain, how "obvious sleep-wake rhythm problems" were excluded? what about chronotypes?

Initial comment:

At first glance, it may appear odd that a sleep duration of 371.0 min (SD 83.0 min) and a sleep efficiency of 66.0% (SD 16.7%), as extracted from the FITBIT, indicates that there is no significant sleep deprivation over the duration of the experiment. Please underline this with appropriate references. 

Answer:

The data recorded by Fitbit showed that sleep duration is enough, but sleep efficiency looks lower than healthy population. Decreased quality of sleep may be attributed to daytime napping opportunity. We added the description to underpin the result (Page 6, Line 245-247).

Follow-up comment:

How the Fitbit algorithm assumes that sleep duration is "enough" (enough compared to what? based on a threshold? is this assumption based on scientific validation? if so, please add the information.

The performed changes in the manuscript (‘Sleep efficiency was relatively lower, presumably because consecutive daytime napping affect night sleep by reducing homeostatic sleep pressure [33] is not suitable and does not anwer the initial concern.

I    Initial comment:

Even though the context is different, no reference is done to studies revealing the impact of habitual daytime napping on night-time sleep and to the literature referring to napping as a health risk indicator - at least in the aged (see also recent publication by Leng et al. 2022). Accordingly, “daytime napping” can, but not necessarily has to be, regarded as a healthy habit. There is indeed a confusing use of the term “habitual napping” (where the volunteers habitual nappers? if so, why?) And its translation to a three-day nap intervention. In the same vain, it is not clear whether the authors believe that napping is beneficial for motor learning performance because of active consolidation processes taking place during nap sleep or because of napping reducing daytime sleepiness/as a countermeasure of sleep deprivation.

      Follow-up comment:

Unless I’m mistaken, the authors did not reply to this comment (neither argue in favor or against of it)

Initial comment:

To my knowledge AASM sleep scoring rules suggest a 30-s window scoring and not a 20s as performed here. Also, I would recommend to report N1 and N2, separately.

      Follow-up comment:

      Unless I’m mistaken, the authors did not reply to this comment.

      Initial comment:

As highlighted by the authors, acquisitions in the nap and awake groups were not performed in the same test environment (at home for the awake group – in the lab for the nap group).  Accordingly, it is mentioned for example in the discussion that the awake group required an overall longer trajectory length compared with the napping group, which may indicate that the groups were not equally focused on the task. Note that if this effect was statistically significant, it needs to be emphasized in the abstract and results sections.

Response:

Your point is precisely the problem, and it was an inevitable limitation of the experimental environment due to the voluntary restraints imposed by COVID-19. 2-way AVOVA revealed a mix of conditions, some with significant main effects and some without statistical differences, including post-hoc. We have stated the description, but it is difficult to ascertain the causal relationship with experimental conditions. We have also added a note to the abstract and results to indicate that there are differences in experimental conditions that should be considered (Page 1, Line 24-25, Page7, Line 270-272).

I fully acknowledge difficulties in experimental research associated with the COVID-19 situation and that procedures had to be adapted accordingly. I agree with the modification suggestion in the abstract, but find the added sentence in the results less clear (“It was noted the difference in the experimental environment: napping group was conducting the RAT task at home, while the control awake group at a laboratory setting”). I would suggest to remove it again at this level as the authors are correct when mentioning in their answer that the respective effects of napping vs no-napping and home vs laboratory environment cannot be levelled out, which is indeed a main concern and makes the data difficult to interpret. However, what about expressing performance as a percentage change from pre- to post-testing? Wouldn’t the assessment of performance change partially account for the confound?

Author Response

Globally, while a series of concerns were addressed in the response letter, other still remain open in my opinion. Rather minor changes have finally been performed in the manuscript. If I'm not mistaken, two of the concerns/comments were not addressed at all in the response to the reviewer document.

Response:

We apologize for not responding to the two points you have raised.

My concerns address the following answers:

Initial comment:

Considering that the authors emphasize that “napping can be regarded as a healthy habit by reducing sleepiness resulting in refreshment in the afternoon, as well as an effective countermeasure to sleep deprivation”, it would have been important to gain information about sleep timing/duration/time since awake prior to the experimental session.

Answer:

... the sleep-wake schedule prior to the start of the experimental intervention was just based on oral instructions, and was not actually measured by EEG or wearable devices. However, the experimenter confirmed the participants' compliance with the conditions prior to the experiment. Therefore, we assume that there were no major disruptions to the participants' sleep wake schedules at least on the day before the experiment. We added the description of the confirmation process in the main text (Page 2, Line 89- Page3, Line 99).

Additionally, not confirming the baseline of sleep-wake schedule prior to the experiment is indeed a deficiency. However, we did not enroll participants with obvious sleep-wake rhythm problems or those with considerably short or long sleep durations in the pre-experiment interviews.

Follow-up comment:

How exactly major disruption or its absence was assessed or quantified? Was there any information (even by questionnaire/interview) about prior sleep duration and timing or any other characteristic? In the same vein, how "obvious sleep-wake rhythm problems" were excluded? what about chronotypes?

Response:

I totally agree with your comment stressing the importance of baseline evaluation prior to the experiment.

We added the detailed description of exclusion criteria in the Method section, instead of the evaluation prior to the experiment (Page 2, Line 89-95). We addressed the score PSQI and MEQ of the participants (Page 3, Line 106-108), which enables us to evaluate the overall quality and circadian preference of the participants. These prescreening would help with the prior evaluation of sleep and circadian rhythm of the participants.

Initial comment:

At first glance, it may appear odd that a sleep duration of 371.0 min (SD 83.0 min) and a sleep efficiency of 66.0% (SD 16.7%), as extracted from the FITBIT, indicates that there is no significant sleep deprivation over the duration of the experiment. Please underline this with appropriate references.

Answer:

The data recorded by Fitbit showed that sleep duration is enough, but sleep efficiency looks lower than healthy population. Decreased quality of sleep may be attributed to daytime napping opportunity. We added the description to underpin the result (Page 6, Line 245-247).

Follow-up comment:

How the Fitbit algorithm assumes that sleep duration is "enough" (enough compared to what? based on a threshold? is this assumption based on scientific validation? if so, please add the information.

The performed changes in the manuscript (‘Sleep efficiency was relatively lower, presumably because consecutive daytime napping affect night sleep by reducing homeostatic sleep pressure [33] is not suitable and does not answer the initial concern.

Response:

Although the detailed algorithm has not appeared to be publicly available, Fitbit would estimate sleep stages using a combination of movement and heart-rate patterns. The data on sleep duration and efficiency (371.0 ± 83.0 min,66.0 ± 16.7%) appear sufficient to show that the participants did not experience obvious sleep deprivation during the experimental period. Again, the point we are trying to make with this data was that no sleep deprivation occurred during the experimental period, and the participants could keep proper sleep habits.

Initial comment:

Even though the context is different, no reference is done to studies revealing the impact of habitual daytime napping on night-time sleep and to the literature referring to napping as a health risk indicator - at least in the aged (see also recent publication by Leng et al. 2022). Accordingly, “daytime napping” can, but not necessarily has to be, regarded as a healthy habit. There is indeed a confusing use of the term “habitual napping” (where the volunteers habitual nappers? if so, why?) And its translation to a three-day nap intervention. In the same vain, it is not clear whether the authors believe that napping is beneficial for motor learning performance because of active consolidation processes taking place during nap sleep or because of napping reducing daytime sleepiness/as a countermeasure of sleep deprivation.

Follow-up comment:

Unless I’m mistaken, the authors did not reply to this comment (neither argue in favor or against of it)

Response:

We are sorry for missing the response. It is essential to ascertain whether they were habitual or non-habitual nappers. The inclusion of habitual and non-habitual nappers would be inappropriate for the study design. In this study, habitual nappers were excluded from the experiment. This criterion was also added to the exclusion criteria (Page 2, Line 89-95). In addition, the unnecessary ‘habitual’ was removed.

Regarding the latter comment, I considered the added sleep-dependent memory consolidation to be primary, rather than the effect of reduced sleepiness. I added these statements to the discussion section (Page 11, Line 371 - Page 12, Line 378).

Initial comment:

To my knowledge AASM sleep scoring rules suggest a 30-s window scoring and not a 20s as performed here. Also, I would recommend to report N1 and N2, separately.

Follow-up comment:

Unless I’m mistaken, the authors did not reply to this comment.

Response:

Again, we apologize for missing the response. Limited to napping, a 20-s window would be allowed for sleep scoring. We added the description of sleep scoring. Also, we revised the description of the sleep stage according to your suggestion (Page 5, Line 199-203).

Initial comment:

As highlighted by the authors, acquisitions in the nap and awake groups were not performed in the same test environment (at home for the awake group – in the lab for the nap group).  Accordingly, it is mentioned for example in the discussion that the awake group required an overall longer trajectory length compared with the napping group, which may indicate that the groups were not equally focused on the task. Note that if this effect was statistically significant, it needs to be emphasized in the abstract and results sections.

Response:

Your point is precisely the problem, and it was an inevitable limitation of the experimental environment due to the voluntary restraints imposed by COVID-19. 2-way AVOVA revealed a mix of conditions, some with significant main effects and some without statistical differences, including post-hoc. We have stated the description, but it is difficult to ascertain the causal relationship with experimental conditions. We have also added a note to the abstract and results to indicate that there are differences in experimental conditions that should be considered (Page 1, Line 24-25, Page7, Line 270-272).

I fully acknowledge difficulties in experimental research associated with the COVID-19 situation and that procedures had to be adapted accordingly. I agree with the modification suggestion in the abstract, but find the added sentence in the results less clear (“It was noted the difference in the experimental environment: napping group was conducting the RAT task at home, while the control awake group at a laboratory setting”). I would suggest to remove it again at this level as the authors are correct when mentioning in their answer that the respective effects of napping vs no-napping and home vs laboratory environment cannot be levelled out, which is indeed a main concern and makes the data difficult to interpret. However, what about expressing performance as a percentage change from pre- to post-testing? Wouldn’t the assessment of performance change partially account for the confound?

Response:

According to your suggestion, the corresponding parts of the Result section were removed.

We calculated a comparison of the percent change between D4 and D11 to offset baseline differences caused by the experimental environment (Page 6, Line 238-240). We added the acquired results in the Result section, respective to the components (Page 8, Line 291-293, Page 10, Line 311-313, Page 11, Line 339-340).

Round 3

Reviewer 1 Report (New Reviewer)

The authors replied to all concerns.

I would suggest:

- to acknowledge the potential impact of chronic napping on circadian sleep regulation when discussing their results.

- verify whether a correlation between sleep stages during recorded naps and performance change could be observed

- verify whether habitual napping is an inclusion or rather an exclusion criterea (in the reply, the authors mention having exlcuded regular nappers, while in the manuscript, habitual nappers are listed as an inclusion criterium, if I'm not wrong.).

p. 2, line 58: "However, whether MA consolidation depends on the manner of sleep remains controversial". I suggest to rephrase as "the manner of sleep" appears rather vague.

p.2, line 68: previous study or studies (i.e. those the authors are referring to before).

p.12, line 374: involved or involving?

Author Response

To Reviewer

Thank you so much for your repeated peer review efforts. We have made the corrections accordingly as noted.

Regarding the description of the behavior meter, we have checked the facts of the experiment and used the abdomen-worn type in this experiment. We have confused it with another experiment and will correct it again. References for validation have also been corrected. (Page 4, Line 152-160)

We used an English proofreading service (Editage®) again to proofread the document in its entirety.

____________________________________________________________________________________

The authors replied to all concerns.

I would suggest:

- to acknowledge the potential impact of chronic napping on circadian sleep regulation when discussing their results.

Response

Thank you for your suggestion. We have incorporated the suggested phrases into the beginning of the discussion. I think the potential effects are a suitable representation of the results of this study. (Page 11, Line 351-352).

- verify whether a correlation between sleep stages during recorded naps and performance change could be observed

Response

While we had tried the correlation analysis, no significant correlations were found by using FDR methods. We stated that no significant correlations analysis were obtained in the manuscript. (Page 6, Line 240-241, Line 257-259)

- verify whether habitual napping is an inclusion or rather an exclusion criterea (in the reply, the authors mention having excluded regular nappers, while in the manuscript, habitual nappers are listed as an inclusion criterium, if I'm not wrong.).

Response

Habitual nappers are excluded in this experiment. We revised the inclusion criteria, which indicated habitual nappers were not enrolled (Page 2, Line 94-95).

  1. 2, line 58: "However, whether MA consolidation depends on the manner of sleep remains controversial". I suggest to rephrase as "the manner of sleep" appears rather vague.

Response

Thank you for your suggestion. We revised the corresponding phrase into “However, whether MA consolidation is mediated by sleep remains controversial”. (Page 2, Line 57-58).

p.2, line 68: previous study or studies (i.e. those the authors are referring to before).

We revised “previous” into “these”. (Page 2, Line 68).

p.12, line 374: involved or involving?

We revised it into “involved”. (Page 12, Line 377).

This manuscript is a resubmission of an earlier submission. The following is a list of the peer review reports and author responses from that submission.

Round 1

Reviewer 1 Report

Authors did the study with aim to explore the effect of daytime napping over multiple days on motor memory learning, more specifically motor adaptation. 

Introduction provide good background information and discuss in detail what this study will add to the literature. Naps has been somewhat controversial about the benefit they add to memory processing and the underlying mechanism that might be implicated, authors have done a great work discussing these issues in the introduction.

Methods are robust, detailed and clearly explained in a reader friendly format.

Results are described in detail, although can be simplified for the readers. The figures should be bigger in size as it was difficult to read the labels and legends in the figures.

Discussion is robust and highlights their finding with incorporating existing data and literature. Limitations are very aptly mentioned and I agree with the authors that there are some nuances which can be improved in the future studies. Nonetheless this article is well written and has very meaningful results.

In conclusion line 423 I believe authors wanted to mention that speed was faster rather than shorter.

Diagram to explain the study design is very helpful for the readers.

I would recommend the authors to simplify presentation of the result section, perhaps first paragraph mentioning what significant results they found followed by the statistical results.

Author Response

Responses to Reviewer #1

Authors did the study with aim to explore the effect of daytime napping over multiple days on motor memory learning, more specifically motor adaptation.

Overall response:

Thank you so much for reviewing our paper. We highly appreciate the generally positive feedback.

As for the statistical analysis, we have corrected the multiple testing as you pointed out. Since there were changes in the results, we have corrected the description of the results and the accompanying discussion section.

We have also made corrections to the figures to reflect the change in the statistics. Since you also pointed out a problem with the image quality, we have replaced it with a figure with high resolution.

Again, thank you for your instructive review.

Introduction: provide good background information and discuss in detail what this study will add to the literature. Naps has been somewhat controversial about the benefit they add to memory processing and the underlying mechanism that might be implicated; authors have done a great work discussing these issues in the introduction.

Response:

Thank you so much for your favorable opinion.

Methods are robust, detailed and clearly explained in a reader friendly format.

Response:

Again, thank you for positive evaluation.

Results are described in detail, although can be simplified for the readers. The figures should be bigger in size as it was difficult to read the labels and legends in the figures.

Response:

We revised the figures with high resolution to help the readers understand clearly.

Discussion is robust and highlights their finding with incorporating existing data and literature. Limitations are very aptly mentioned and I agree with the authors that there are some nuances which can be improved in the future studies. Nonetheless this article is well written and has very meaningful results.

Response:

We highly appreciate your thought.

In conclusion line 423 I believe authors wanted to mention that speed was faster rather than shorter.

Response:

We replaced with “faster” according to your suggestion.

Diagram to explain the study design is very helpful for the readers.

Response:

Thank you for recognizing the value of the diagram.

I would recommend the authors to simplify presentation of the result section, perhaps first paragraph mentioning what significant results they found followed by the statistical results.

Response:

Thank you for instructional suggestion. As mentioned at the beginning of this report, other reviewers pointed out the issues of multiple testing, so the results have also been corrected.

We think it is more concise than the first draft.

---------------------------------------------------------------------------------

Reviewer 2 Report

The current study investigated how taking a daytime nap on consecutive days influences motor learning. The authors concluded that daytime napping significantly improved speed and accuracy of a motor learning task. However, the data do not seem to support the authors’ conclusion.

There were three measures: required time, trajectory length, and synthetic index. The main effect of measurement point was significant in all three measures, but it could be simply due to the practice or learning effect in both the napping and wake groups.

Across all three measures no significant main effect of group (napping vs. awake) was found. More importantly, there was no significant interaction between group and measurement point in all three measures. Significant interaction was not found even when the data were divided by each angle.

The only significant effect was found in the final test session (RT session) in the synthetic index (only in the 30 degree angle condition). Nevertheless, different performance in the last session does not necessarily mean improved performance when there is no significant interaction.

There were some significant differences between the napping and wake groups in some of the measurement points. However, the authors did not report statistical analyses on each measurement point (just depicted asterisks and cross symbols on the graphs). Also, it is not clear these analyses on measurement points were corrected for multiple comparisons.

On a minor point, the authors should consider the quality of figures. It is difficult to read the graphs with low resolution.

Author Response

Responses to Reviewer #2

Overall response:

Thank you so much for reviewing our paper. We highly appreciate the generally positive feedback.

As for the statistical analysis, we have corrected the multiple testing as you pointed out. Since there were changes in the results, we have corrected the description of the results and the accompanying discussion section.

We have also made corrections to the figures to reflect the change in the statistics. Since you also pointed out a problem with the image quality, we have replaced it with a figure with high resolution.

Again, thank you for your instructive review.

The current study investigated how taking a daytime nap on consecutive days influences motor learning. The authors concluded that daytime napping significantly improved speed and accuracy of a motor learning task. However, the data do not seem to support the authors’ conclusion.

There were three measures: required time, trajectory length, and synthetic index. The main effect of measurement point was significant in all three measures, but it could be simply due to the practice or learning effect in both the napping and wake groups.

Response:

We agree with your point that the significant difference in retest important, not the results of the last session. I have corrected the description of the results. It is true that the improvement trend in the last session can be attributed to the practice effect. However, the retest result one week after the end of the last session is not a result of practice, but rather an effect of offline fixation. This also overlaps with the answer to the following question, but we have revised the description and the argument.

Across all three measures no significant main effect of group (napping vs. awake) was found. More importantly, there was no significant interaction between group and measurement point in all three measures. Significant interaction was not found even when the data were divided by each angle.

The only significant effect was found in the final test session (RT session) in the synthetic index (only in the 30 degree angle condition). Nevertheless, different performance in the last session does not necessarily mean improved performance when there is no significant interaction.

Response:

We agree with your point that the significant difference in retest important, not the results of the last session. I have corrected the description of the results.

There were some significant differences between the napping and wake groups in some of the measurement points. However, the authors did not report statistical analyses on each measurement point (just depicted asterisks and cross symbols on the graphs). Also, it is not clear these analyses on measurement points were corrected for multiple comparisons.

Response:

Thank you for suggestion for multiple comparison. We added the descriptions of methods and results for multiple testing to the respective sections.

On a minor point, the authors should consider the quality of figures. It is difficult to read the graphs with low resolution.

Response:

As noted at the outset, we prepared the figures with high resolution.

---------------------------------------------------------------------------------

Round 2

Reviewer 2 Report

The authors have made significant changes in the revised manuscript. However, the results are not yet convincing enough.

  1. Time data showed that the wake group’s performance was worse on Day 11 compared to the nap group, only in the 30 degrees condition. The authors argued that this difference implies that daytime napping improved motor learning. However, there is a possibility that the nap and wake groups were different from the beginning. There were significant effects of group in the results from other measures (distance & synthetic index) but interaction between measurement points and group was not significant. If the nap and wake groups were heterogeneous from Day 1-Pre, then the significant difference observed in the Time measures at 30 degrees might not mean memory consolidation.
  1. The authors explained that the FDR method was applied to ANOVA p-values. However, it is not clear whether any corrections were applied to post-hoc t-tests. The only significant effect was found on Day 11, at 30 degrees in the Time data. Though the effect size (d) was quite large, this effect could disappear if the p-value (.034) is corrected for multiple comparisons.